# Identification of Therapeutic Targets in Autism Spectrum Disorder through CHD8-Notch Pathway Interaction Analysis

**Hewei Zhang[1,2]⊚, Shenghao Hua[2]⊚, Daiyan Jiao[1,3], Dong Chen[1], Qin Gu[2], Chao Bao[1]\***

**1** The Affiliated Hospital of Nanjing University of Chinese Medicine, Jiangsu Province Hospital of Chinese Medicine, the First Clinical Medical College, Nanjing, Jiangsu, China, **2** The Children's Hospital of Soochow University, Suzhou, Jiangsu, China, **3** Haian Hospital affiliated to Nantong University, Hai'an, Jiangsu, China.

⊚ These authors contributed equally to this work.
\* drbaochao@163.com

## Abstract

### Background

Autism Spectrum Disorder (ASD) is a complex neurodevelopmental disorder with a rising global prevalence. Mutations in the CHD8 gene have been implicated in ASD, yet the underlying molecular mechanisms remain insufficiently understood.

### Methods

We analyzed transcriptomic data from the CHD8A and CHD8B allelic deletion sample dataset GSE236993 to identify differentially expressed genes (DEGs). We intersected these DEGs with genes related to the Notch signaling pathway and performed functional enrichment analyses, including Gene Ontology (GO) and Kyoto Encyclopedia of Genes and Genomes (KEGG) pathway enrichment, as well as protein-protein interaction (PPI) analyses, to identify key genes. These key genes were validated using the CHD8-deficient sample dataset GSE85417, resulting in the identification of seven common key genes. We then constructed drug-gene interaction networks and microRNA (miRNA) regulatory networks to further elucidate the mechanisms by which CHD8 impacts ASD.

### Results

Seven hub genes—IGF2, FN1, CXCR4, COL11A1, ITGA6, LOX, and FBN2—were identified, all involved in the Notch signaling pathway and playing significant roles in neurodevelopment and extracellular matrix regulation. Among these, IGF2 and CXCR4 were particularly crucial in ASD pathogenesis, suggesting their potential as diagnostic biomarkers and therapeutic targets. MiRNA regulatory network analysis revealed several miRNAs that may modulate these hub genes, offering new insights

**Data availability statement:** All relevant data are within the manuscript and its Supporting Information files.

**Funding:** The author(s) received no specific funding for this work.

**Competing interests:** The authors have declared that no competing interests exist.

into ASD pathogenesis. Drug-gene interaction analysis suggested possible therapeutic small-molecule compounds, such as AMD3100 and IGF-1R inhibitors.

## Conclusions

Our multi-level bioinformatics analysis identified key genes and regulatory networks potentially involved in ASD associated with CHD8 deficiency. These findings enhance the understanding of ASD's molecular mechanisms and highlight potential therapeutic targets, paving the way for future diagnostic and treatment strategies.

## Introduction

Autism spectrum disorder (ASD) is a complex neurodevelopmental condition with a rising global prevalence, currently affecting 1% to 3% of children worldwide [1,2]. The increasing incidence of ASD presents substantial health and economic challenges for both society and families. In China, similar trends are observed, with epidemiological studies reporting a growing prevalence of ASD [3]. ASD is characterized by considerable phenotypic heterogeneity, primarily manifesting as impairments in social communication, restricted interests, and repetitive behaviors [4]. Given the intricate interactions between genetic and environmental factors in the etiology of ASD [5,6], uncovering the underlying pathophysiological mechanisms is essential for improving diagnostic and therapeutic approaches.

Among the genetic factors implicated in ASD, mutations in the CHD8 gene are frequently reported [7,8]. CHD8 is a key transcriptional regulator that plays a crucial role in gene expression during neurodevelopment [9,10]. Loss or mutation of CHD8 is associated with abnormal neuronal development, resulting in hallmark ASD features [11,12]. CHD8 is recognized as a high-risk gene for ASD and is involved in regulating multiple signaling pathways [13–15]. However, the precise molecular mechanisms through which CHD8 contributes to ASD pathogenesis, particularly its role in specific signaling pathways, remain incompletely understood.

Recent evidence has suggested that CHD8 may directly or indirectly regulate components of the Notch signaling pathway, which is known to influence neural progenitor cell proliferation and differentiation. For example, CHD8 knockdown alters expression of Notch-related genes in human neural progenitor cells, as demonstrated by ChIP-seq analyses [16]. Additionally, CHD8-dependent chromatin remodeling affects neurodevelopmental pathways that overlap with Notch signaling [10,14]. These findings provide a rationale for exploring CHD8–Notch interactions in the context of ASD pathogenesis.

The Notch signaling pathway is central to neurodevelopment, influencing processes such as neuronal differentiation, cell fate determination, and synapse formation [17–19]. Dysregulation of the Notch pathway has been implicated in the etiology of ASD, potentially contributing to abnormal neuronal development and the cognitive and behavioral deficits observed in affected individuals [20,21]. Despite growing interest in the role of Notch signaling in ASD, its direct interaction with CHD8 remains largely unexplored.

To investigate the interaction between CHD8 and the Notch signaling pathway in ASD, we analyzed differentially expressed genes (DEGs) in CHD8-deficient samples and identified genes that intersect with the Notch pathway. We conducted bioinformatics analyses, including Gene Ontology (GO) and Kyoto Encyclopedia of Genes and Genomes (KEGG) pathway enrichment, to elucidate the biological functions of these intersecting genes in neurodevelopment. Additionally, we constructed protein-protein interaction (PPI) networks, microRNA (miRNA) regulatory networks, and drug-gene interaction networks to comprehensively assess the impact of CHD8 deficiency on the Notch pathway and its downstream gene networks. These analyses aim to provide novel insights into the molecular mechanisms underlying ASD and identify potential therapeutic targets.

## Materials and methods

### Microarray data acquisition and normalization

We searched the GEO database using the keywords "CHD8" [All Fields] OR "autism" [All Fields] to identify gene expression datasets related to autism and CHD8. The inclusion criteria were: (1) studies focusing on CHD8 gene-related functional research; (2) samples comprising brain tissue or neuronal cells; and (3) species including humans, mice, or rats. Based on these criteria, datasets GSE236993 and GSE85417 were selected for analysis. Both datasets contain transcriptome data from CHD8 allelic deletion samples. Data normalization was performed using the "limma" package in R to ensure comparability across different samples.

### Identification of DEGs

DEGs between CHD8-deficient (CHD8A and CHD8B) and wild-type samples were identified using the "limma" package in R, which employs linear models and empirical Bayes methods.To control for multiple comparisons and reduce false positives, adjusted p-values were calculated using the Benjamini-Hochberg False Discovery Rate (FDR) method. Genes with an adjusted p-value (FDR) < 0.05 and $|\log_2$ fold change$| > 1$ were defined as DEGs.. The DEGs were visualized using volcano plots and heatmaps generated with the "ggplot2" and "pheatmap" packages, respectively, to provide intuitive representations of their distribution and expression patterns.

### Functional and pathway enrichment analysis

Two gene lists were generated based on the DEGs identified from CHD8A and CHD8B samples. Intersection analysis was performed using the "VennDiagram" package to identify common DEGs between the CHD8A and CHD8B deletion samples, with Venn diagrams constructed for visualization. Subsequently, GO and KEGG pathway enrichment analyses were performed on the DEGs using the "clusterProfiler" and "enrichplot" packages. The GO analysis included Biological Process (BP), Molecular Function (MF), and Cellular Component (CC) categories, with significant results determined by an adjusted p-value less than 0.05. For each category, the top 10 enriched pathways were visualized.

### PPI network construction and hub gene identification

A PPI network of the DEGs was constructed using the STRING database (http://string-db.org/) with a confidence score threshold of 0.4. The PPI network was visualized using Cytoscape software. The cytoHubba plugin in Cytoscape was employed to further analyze the PPI network, identifying hub genes with significant connectivity.

### Construction of drug-DEG interaction network

The potential interactions between hub genes and drugs or small molecules were investigated using the Drug-Gene Interaction Database (DGIdb, https://www.dgidb.org). The identified drug-gene interactions were visualized using Cytoscape to construct a drug-DEG interaction network, providing insights into potential therapeutic target development.

### Hub Gene–miRNA interaction network

A hub gene–microRNA (miRNA) interaction network was constructed using the miRWalk database (http://mirwalk.umm.uni-heidelberg.de/), which contains both predicted and experimentally validated miRNA-target interactions. miRNAs with a score greater than 0.95 and associated with at least four hub genes were selected. The final miRNA–hub gene interaction network was visualized in Cytoscape, revealing the potential roles of miRNAs in regulating these critical genes.

### Statistical analysis

All statistical analyses were performed utilizing the R statistical software (version 4.2.1, http://r-project.org/). The independent t-test was employed for continuous data that followed a normal distribution, while the Mann-Whitney U test was applied to continuous data that did not conform to normality. All statistical analyses were two-tailed, with a significance threshold set at $P \leq 0.05$."

## Results

### Identification of differentially expressed genes

To investigate the effects of CHD8 deficiency, we analyzed the GSE236993 dataset to identify differentially expressed genes (DEGs) between wild-type samples and those with CHD8A or CHD8B allelic deletions. Volcano plots illustrated significant upregulation and downregulation of genes in both groups. Specifically, in the CHD8A group, 672 genes were upregulated and 1206 genes were downregulated (Fig 1A), while in the CHD8B group, 499 genes were upregulated and 1061 genes were downregulated (Fig 1B). Venn diagram analysis revealed that 1120 DEGs were shared between the CHD8A and CHD8B groups, and 298 of these genes were primarily associated with the Notch signaling pathway (Fig 1C). Heatmap analysis further visualized differences in gene expression patterns between wild-type and CHD8-deficient samples, clearly showing trends of upregulated and downregulated genes (Fig 1D). Additionally, Supplementary Table S1 in S1 File summarizes the top 20 DEGs ranked by significance, including their gene symbols, $\log_2$ fold changes, adjusted p-values (FDR), and functional annotations.

### Functional and pathway enrichment analysis

GO enrichment and KEGG pathway analyses were performed on the 298 shared DEGs, revealing several key biological processes and pathways associated with ASD (Fig 2A). GO analysis indicated significant enrichment in biological processes such as regulation of cell development, nervous system development, neurogenesis, and the Notch signaling pathway, highlighting the potential roles of these genes in neurodevelopment. KEGG pathway analysis identified multiple pathways relevant to ASD, including the PI3K-Akt signaling pathway, TGF-β signaling pathway, and ECM–receptor interaction, which play critical roles in neurodevelopment and extracellular matrix regulation (Fig 2B).

### PPI network construction and hub gene identification

A PPI network was constructed using the identified DEGs, leading to the identification of several hub genes, including NOTCH1, FN1, BDNF, and PAX6, which may play critical roles in the molecular mechanisms of ASD (Fig 3A). Further network analysis revealed complex interactions among these hub genes, suggesting they may influence ASD pathogenesis through a shared regulatory network (Fig 3B).

### Validation of hub gene expression

To validate the hub genes identified from the interaction network, we incorporated an additional transcriptome dataset (GSE85417) involving CHD8-deficient samples. Differentially expressed genes were identified between CHD8-deficient and wild-type samples, with 197 genes significantly upregulated and 336 genes significantly downregulated, as illustrated

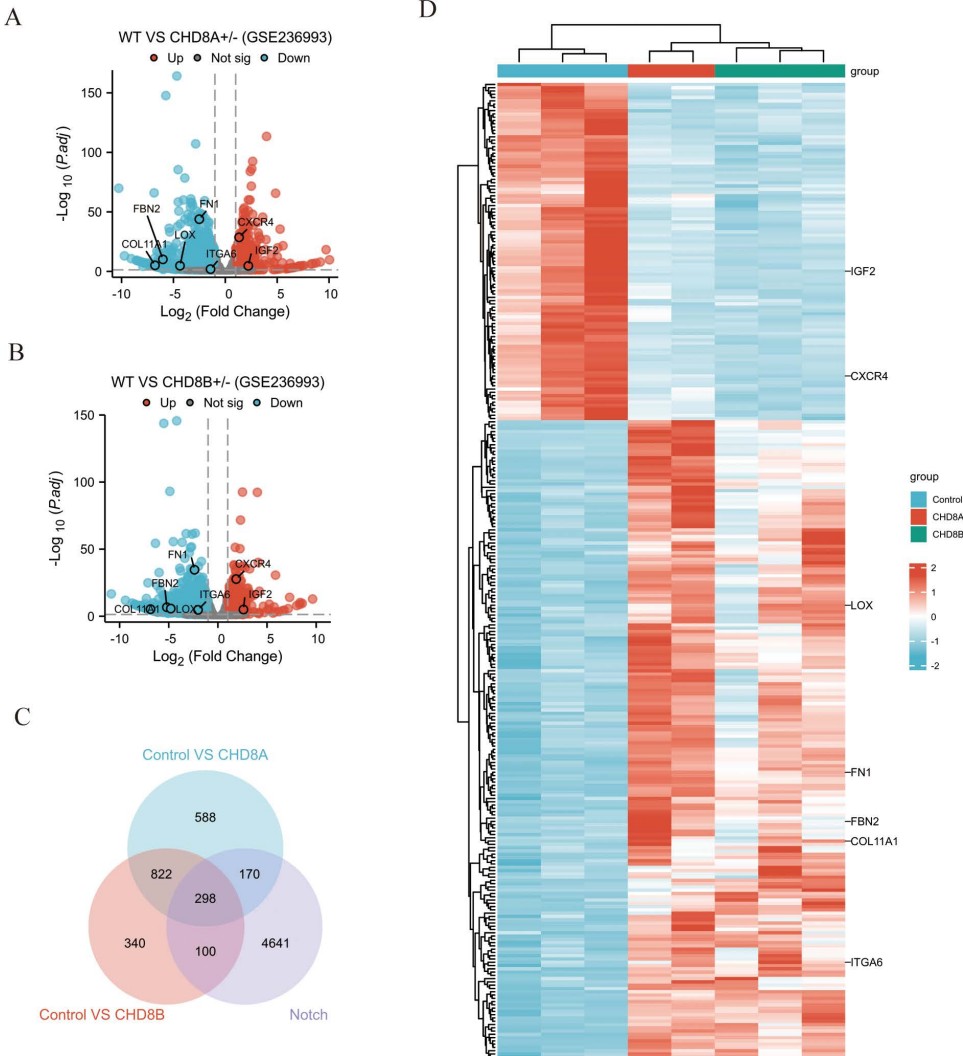

**Fig 1. Identification of DEGs associated with CHD8 deficiency.** (A) Volcano plot of DEGs between wild-type (WT) and CHD8A+/– samples (GSE236993). Red and blue dots represent significantly up- and downregulated genes, respectively; gray indicates non-significant changes. (B) Volcano plot of DEGs between WT and CHD8B+/– samples (GSE236993), as in (A). (C) Venn diagram showing the overlap of DEGs from CHD8A and CHD8B groups with Notch pathway-related genes. A total of 298 genes are primarily linked to Notch signaling. (D) Heatmap depicting expression patterns of the identified DEGs in WT and CHD8-deficient samples. Red represents upregulation, blue represents downregulation.

by the volcano plot (Fig 4A). Venn diagram analysis comparing the DEGs from GSE236993 with the key genes from GSE85417 identified seven common genes (Fig 4B). This finding further supports the potential importance of these genes in the regulation of the Notch signaling pathway and their involvement in the molecular mechanisms of ASD associated with CHD8 deficiency. Heatmap visualization of the expression changes of these seven common genes between CHD8-deficient and wild-type samples demonstrated that LOX, COL11A1, and ITGA6 were significantly upregulated, while FBN2 and CXCR4 were downregulated in CHD8-deficient samples (Fig 4C). These results suggest that these seven genes may play pivotal roles in the molecular mechanisms triggered by CHD8 deficiency and are critical in ASD pathogenesis.

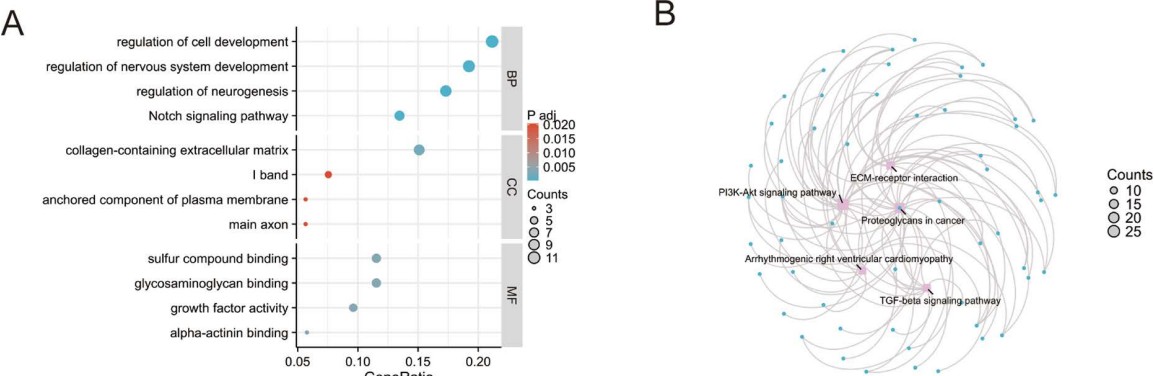

**Fig 2. GO and KEGG enrichment analyses of shared DEGs.** (A) GO enrichment results showing significant involvement in cell development, nervous system development, neurogenesis, and the Notch signaling pathway. (B) KEGG pathway enrichment highlighting pathways such as PI3K-Akt, TGF-β, and ECM–receptor interaction.

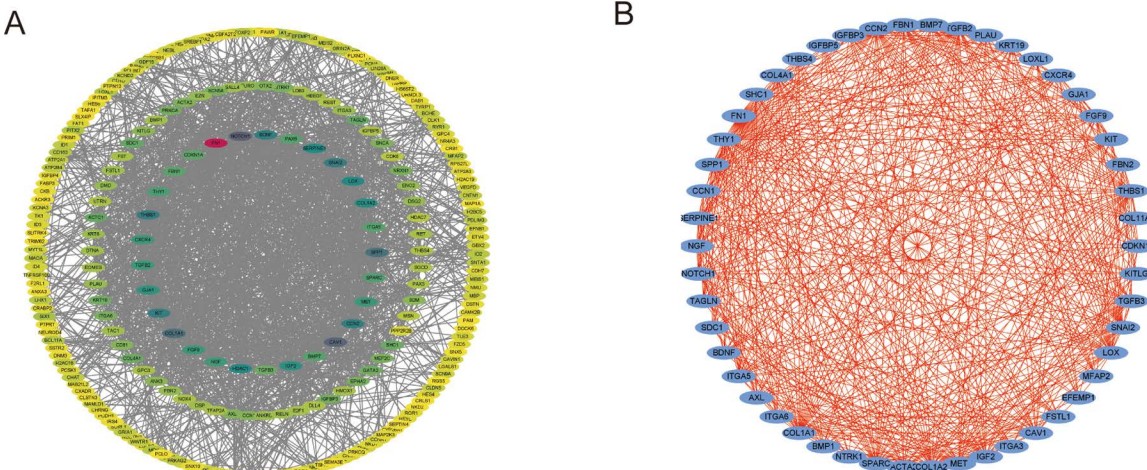

**Fig 3. PPI network and hub gene analysis.** (A) PPI network of identified DEGs, highlighting hub genes such as NOTCH1, FN1, BDNF, and PAX6. (B) Expanded network showing complex interactions among hub genes, indicating a shared regulatory network involved in ASD pathogenesis.

## Drug-DEG interaction network

The drug–DEG interaction network analysis revealed potential interactions between key genes associated with CHD8A and CHD8B allelic deletions—such as FN1, COL11A1, CXCR4, IGF2, and FBN2—and several small-molecule compounds. Fig 5 illustrates the relationships between these genes and potential therapeutic compounds, indicating that these small molecules may modulate the expression or function of the DEGs, suggesting potential avenues for therapeutic intervention.

## miRNA–Hub Gene Regulatory Network

The miRNA–hub gene interaction network analysis revealed regulatory relationships between several key genes and their corresponding microRNAs (miRNAs). Fig 6 shows that hub genes such as IGF2, COL11A1, CXCR4, ITGA6, LOX, FBN2,

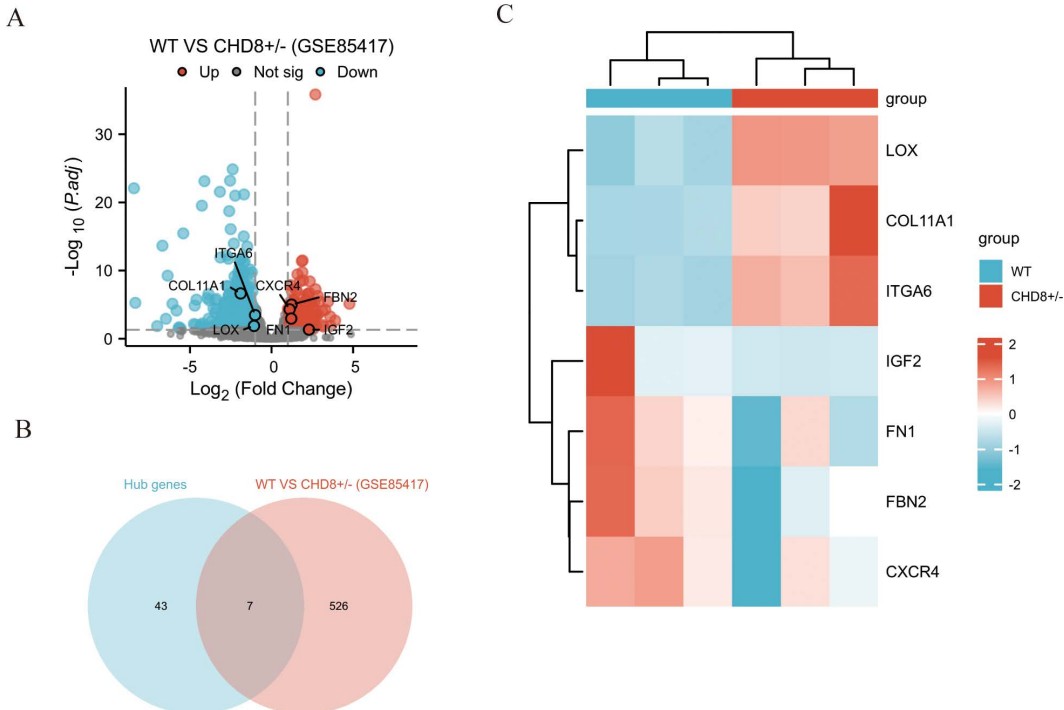

**Fig 4. Validation of hub genes in GSE85417.** (A) Volcano plot of DEGs between wild-type and CHD8-deficient samples from GSE85417. (B) Venn diagram showing seven genes overlapping between GSE236993-derived key genes and GSE85417 DEGs. (C) Heatmap illustrating expression changes of the seven common genes, with LOX, COL11A1, and ITGA6 upregulated, and FBN2 and CXCR4 downregulated in CHD8-deficient samples, supporting their critical roles in ASD pathogenesis.

and FN1 are regulated by multiple miRNAs. These miRNAs are likely to play important regulatory roles in ASD, indicating their potential as therapeutic targets.

## Discussion

ASD is a prevalent neurodevelopmental disorder that imposes substantial health and economic burdens on affected individuals and their families [22]. In this study, we performed an in-depth analysis of a Gene Expression Omnibus (GEO) dataset to identify differentially expressed genes (DEGs) in samples with CHD8A and CHD8B allelic deletions. Our analysis revealed several key DEGs, which were further subjected to GO enrichment and KEGG pathway analyses.Compared to previous studies that have broadly investigated CHD8 or Notch signaling in neurodevelopmental disorders, our work presents several key innovations. First, we conducted a focused integrative analysis of two independent transcriptomic datasets (GSE236993 and GSE85417), both involving CHD8 allelic deletions, to identify reproducible and robust gene expression changes. Second, our analysis emphasizes the intersection between CHD8-regulated genes and the Notch signaling pathway, a convergence that has been underexplored in ASD research. By integrating differential gene expression, pathway enrichment, protein–protein interaction, drug–gene interaction, and miRNA regulatory networks, we constructed a comprehensive regulatory landscape that highlights specific hub genes such as *IGF2*, *CXCR4*, and *FN1* as potential therapeutic targets. These methodological advances and findings extend beyond prior studies by identifying clinically actionable targets within a defined molecular axis (CHD8–Notch) and by providing a multi-level framework for ASD biomarker discovery and future intervention development.

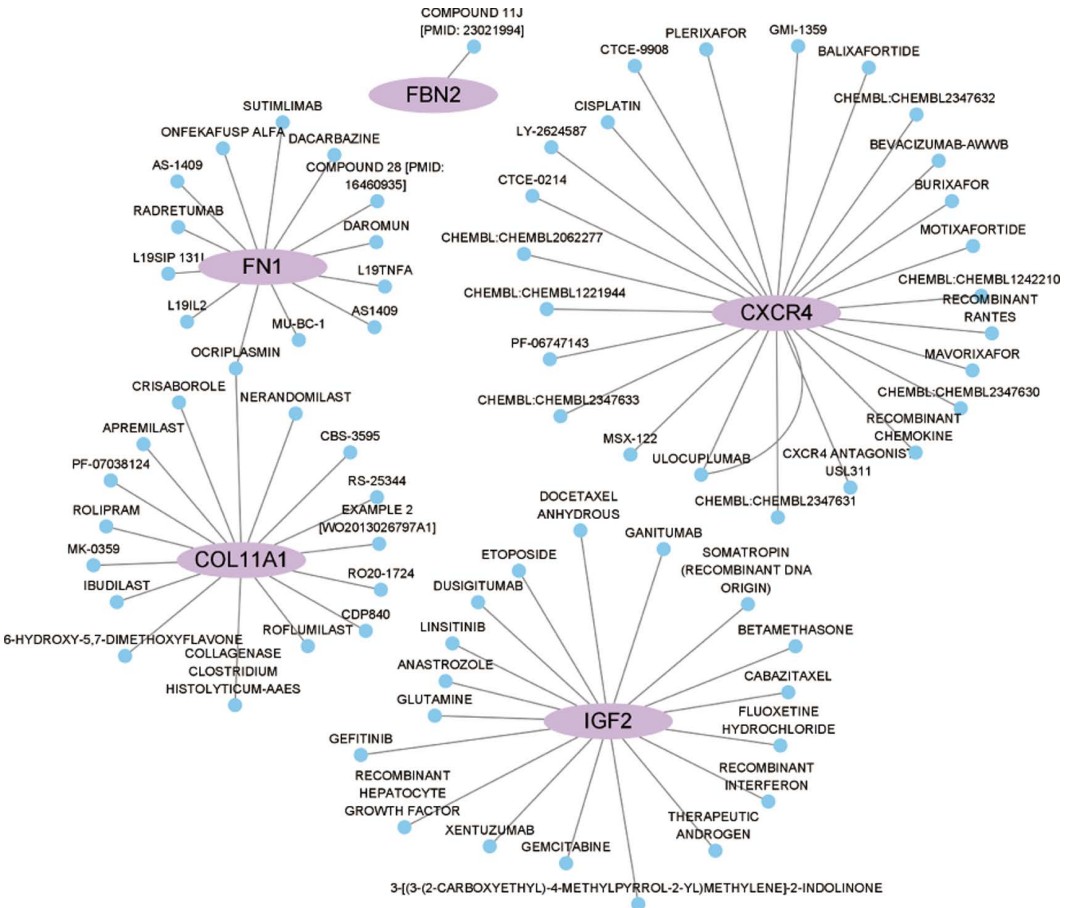

**Fig 5. Drug–gene interaction network.** Key genes (FN1, COL11A1, CXCR4, IGF2, FBN2) and their potential small-molecule modulators are shown, suggesting possible therapeutic interventions for ASD.

GO analysis indicated that these DEGs were significantly enriched in biological processes related to nervous system development, neuronal migration, synaptic transmission, and the Notch signaling pathway, suggesting their crucial roles in the development and function of the nervous system. KEGG pathway analysis identified several pathways that were significantly enriched and are known to be associated with ASD, including the PI3K-Akt signaling pathway, extracellular matrix (ECM)–receptor interaction, and the transforming growth factor-beta (TGF-β) signaling pathway. These pathways play critical roles in neurodevelopment, ECM regulation, and neuronal signaling, highlighting their potential involvement in the pathophysiology of ASD.

Using the GSE85417 dataset for validation, we identified seven key genes within the Notch signaling pathway affected by CHD8 deficiency: LOX, COL11A1, CXCR4, IGF2, ITGA6, FN1, and FBN2. LOX and COL11A1 are involved in extracellular matrix (ECM) formation and remodeling, which may influence brain development by altering the neuronal microenvironment. CXCR4 and IGF2 play critical roles in neuronal migration and differentiation, and their dysregulation has been linked to abnormal neurodevelopment associated with Autism Spectrum Disorder (ASD). ITGA6, FN1, and FBN2 are important for cell adhesion, signal transduction, and maintenance of tissue structure; their dysfunction could lead to impaired neuronal connectivity, affecting behavioral and cognitive functions in individuals with ASD.. In particular, AMD3100, a CXCR4 antagonist, has been shown to inhibit aberrant CXCR4 signaling, which is implicated in abnormal

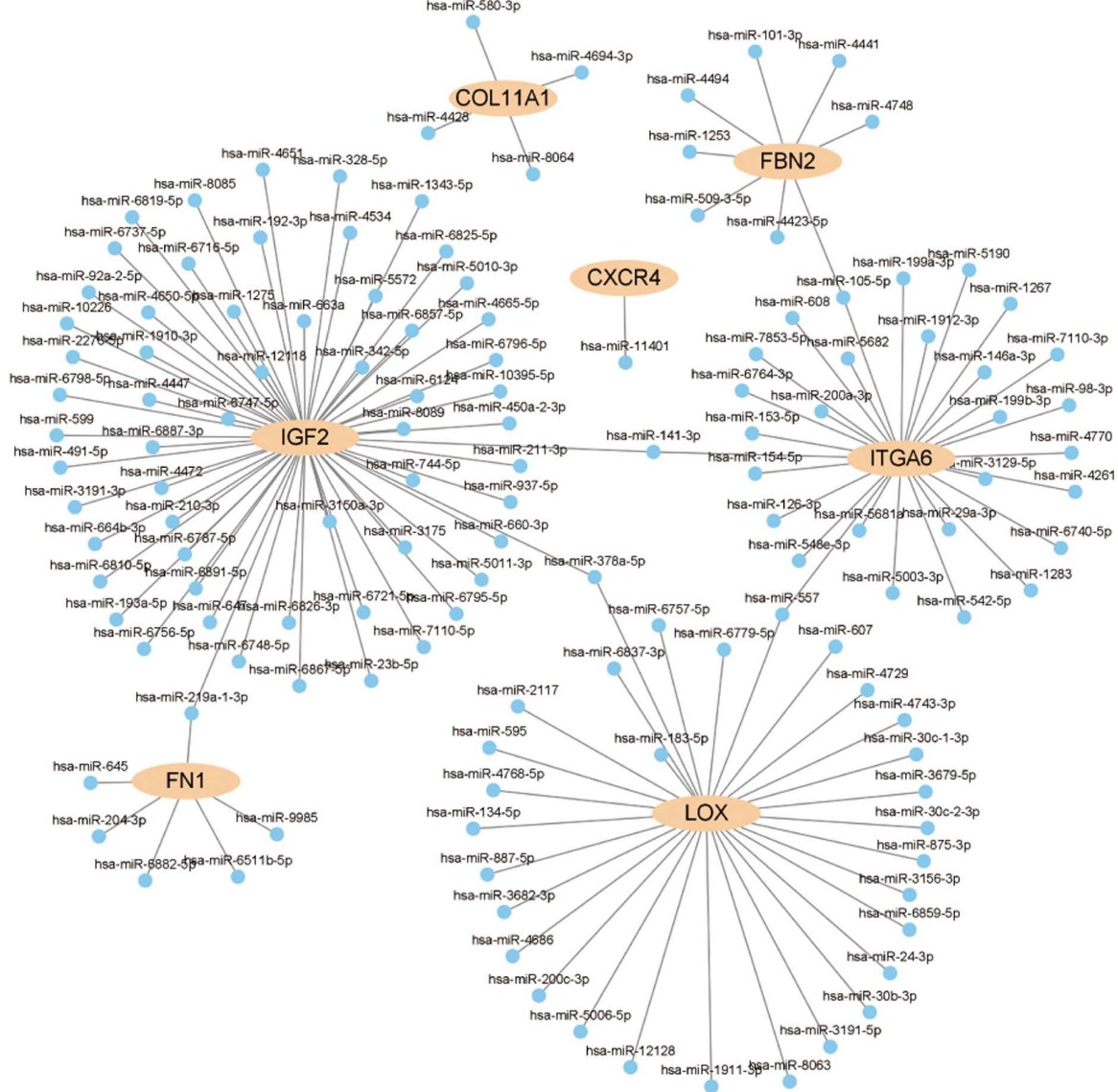

**Fig 6. miRNA–hub gene interaction network.** Identified hub genes (IGF2, COL11A1, CXCR4, ITGA6, LOX, FBN2, FN1) are regulated by multiple miRNAs, suggesting important roles in ASD and potential therapeutic targets.

neuronal migration and cortical development in ASD models. Preclinical studies have demonstrated that CXCR4 inhibition can normalize excitatory/inhibitory balance and improve social behaviors in mouse models of autism [20]. Similarly, IGF-2 inhibitors target the insulin-like growth factor signaling pathway, which plays a key role in synaptic maturation and plasticity. Modulation of IGF-2 activity has shown promise in correcting synaptic dysfunction and improving cognitive function in ASD animal models and in clinical trials involving related neurodevelopmental disorders such as Rett syndrome [23]. These findings provide preliminary evidence that targeting CXCR4 and IGF-2 may offer therapeutic benefits for individuals with ASD, particularly those exhibiting CHD8-related molecular alterations. However, further experimental and clinical validation is required to evaluate the efficacy and safety of these compounds in ASD populations. Overall, these genes not

only play significant roles in neurodevelopment and ECM regulation but also hold potential as diagnostic and therapeutic targets for ASD.

Among the seven key genes identified, several have well-established roles in neural development and show potential connections to Notch signaling. IGF2 is a growth factor involved in brain development, synaptic plasticity, and neurogenesis, and may modulate Notch signaling through regulation of downstream proliferative cues. CXCR4, a chemokine receptor, has been linked to neuronal migration and axonal guidance; its cross-talk with Notch signaling contributes to stem cell maintenance and neuroprogenitor positioning during development. FN1 (Fibronectin 1) is essential for cell adhesion and ECM structure, and has been shown to influence Notch ligand presentation and activation. COL11A1 and FBN2 are ECM-related proteins that may indirectly regulate Notch signaling by modifying the cellular microenvironment. ITGA6, a laminin-binding integrin, plays a role in radial glial scaffold formation and neuronal migration, potentially interacting with Notch-related cell fate pathways. LOX (Lysyl oxidase) contributes to ECM remodeling and may influence Notch pathway activation by altering ligand accessibility or mechanosensory contexts. Collectively, these genes may converge on neurodevelopmental processes such as synaptogenesis, axon pathfinding, and cortical organization, key domains disrupted in ASD.

The analysis of the hub gene–microRNA (miRNA) interaction network further revealed complex regulatory relationships involving these key genes and their corresponding miRNAs. Genes such as IGF2, CXCR4, and FN1 are regulated by multiple miRNAs, highlighting the critical role of miRNAs in fine-tuning gene expression. These miRNAs may regulate the expression levels of key genes, impacting neuronal development, differentiation, and ECM modulation, thereby playing an essential role in the pathogenesis of ASD. These findings suggest that specific miRNAs may serve as potential therapeutic targets in ASD, providing a foundation for the future development of miRNA-based therapeutic strategies.

However, our study has several limitations. First, due to the limited availability of transcriptomic data from human ASD samples, the majority of our data were derived from animal models and cell lines, which may limit the generalizability of our findings to human ASD. Second, because our analysis was based on microarray data without inclusion of RNA sequencing data, we may have incompletely detected significant gene expression changes. Furthermore, although we validated the transcriptional changes of the hub genes, comprehensive validation at the protein level is lacking, which we aim to address in future studies.

Furthermore, we acknowledge the lack of experimental validation using in vivo or in vitro models as a limitation of our study. To address this, we plan to conduct future experiments to validate the expression levels of key genes, such as *IGF2* and *CXCR4*, in patient-derived samples and neuronal cell models. These experiments will help confirm the bioinformatic findings and provide deeper insights into the mechanistic roles of these genes in ASD pathogenesis.

In conclusion, through multi-level bioinformatics analyses, this study elucidated the crucial roles of CHD8-related genes in the pathogenesis of ASD, identifying potential therapeutic targets and regulatory miRNAs. These findings provide valuable insights into the molecular mechanisms of ASD and offer important directions for future drug development and targeted therapies.

## Conclusion

In this study, through the analysis of CHD8A and CHD8B allelic deletion samples, we identified seven hub genes associated with Autism Spectrum Disorder (ASD): IGF2, FN1, CXCR4, COL11A1, ITGA6, LOX, and FBN2. We conducted an in-depth analysis of the associated signaling pathways, miRNA regulatory networks, and potential small-molecule compounds targeting these hub genes. These key genes play significant roles in neurodevelopment and extracellular matrix regulation, with IGF2 and CXCR4 likely playing pivotal roles in the pathogenesis of ASD. Therefore, these hub genes, particularly IGF2 and CXCR4, hold promise as potential diagnostic biomarkers and therapeutic targets for ASD. Moreover, the identification of associated miRNAs provides new avenues for developing innovative therapeutic strategies.

However, the precise roles of these hub genes and their regulatory miRNAs in ASD require further investigation to elucidate their underlying mechanisms, which could pave the way for more effective therapeutic approaches. This study offers valuable insights into the molecular underpinnings of ASD and highlights several promising candidates for future targeted therapies and interventions.

## Supporting information

**S1 File. Supplementary Table S1.** This file includes: The top 20 differentially expressed genes (DEGs), including their log2 fold changes, p-values, and gene annotations. Actual gene counts or gene ratios involved in each enriched Gene Ontology (GO) term pathway. A detailed description of the processing and analysis procedures for the GSE236993 and GSE85417 datasets.
(DOCX)

## Author contributions

**Formal analysis:** Daiyan Jiao.

**Methodology:** Dong Chen.

**Supervision:** chao bao.

**Writing – original draft:** Hewei Zhang, Shenghao Hua.

**Writing – review & editing:** Qin Gu, chao bao.

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
