## [Decision Letter · Decision Letter 0]

Dear Dr. bao,

Thank you for submitting your manuscript to PLOS ONE. After careful consideration, we feel that it has merit but does not fully meet PLOS ONE’s publication criteria as it currently stands. Therefore, we invite you to submit a revised version of the manuscript that addresses the points raised during the review process.

We look forward to receiving your revised manuscript.

Kind regards,

Jinhui Liu

Academic Editor

PLOS ONE

Additional Editor Comments:

Authors should revise according to the suggestions of reviewers. The modifications should be marked. A point to point response letter is needed.

Reviewers' comments:

Reviewer's Responses to Questions

**Comments to the Author**

1. Is the manuscript technically sound, and do the data support the conclusions?

Reviewer #1: Yes

Reviewer #2: Yes

2. Has the statistical analysis been performed appropriately and rigorously?

Reviewer #1: Yes

Reviewer #2: Yes

3. Have the authors made all data underlying the findings in their manuscript fully available?

Reviewer #1: Yes

Reviewer #2: Yes

4. Is the manuscript presented in an intelligible fashion and written in standard English?

Reviewer #1: Yes

Reviewer #2: Yes

Reviewer #1: This study investigates the molecular mechanisms underlying Autism Spectrum Disorder (ASD) by exploring the interaction between CHD8—a high-risk gene in ASD—and the Notch signaling pathway. Using publicly available transcriptomic datasets, the authors performed a comprehensive bioinformatics analysis including differential expression analysis, enrichment analysis, protein-protein interaction (PPI) network construction, and regulatory network exploration via miRNAs and drug-gene interaction databases. Seven potential hub genes were identified, with IGF2 and CXCR4 highlighted as key players in ASD pathogenesis. The study is scientifically meaningful and methodologically sound, but several major issues need to be addressed before publication.

1.Please elaborate on why the Notch signaling pathway was prioritized for interaction analysis with CHD8. Citing studies that show CHD8 indirectly or directly regulates Notch components would support this choice.

2.Clarify whether adjusted p-values (FDR) were used to define DEGs. If not, please re-analyze using adjusted values to reduce false positives. Include a summary table of the top 20 DEGs with their log2 fold change, p-value, and annotation.

3. Add labels for representative genes on the volcano plots (e.g., IGF2, FN1, CXCR4). Include color scales and clustering dendrograms in the heatmaps for better interpretability.

4. In Figure 2, please provide actual gene counts or ratios involved in each GO term/pathway.

5.While several small-molecule compounds (e.g., AMD3100 and IGF-1R inhibitors) are identified, the therapeutic implications for ASD are not well explained.

6.The manuscript is generally understandable, but contains grammatical and stylistic issues. Consider professional English language editing to improve clarity and scientific tone.

Reviewer #2: Overall, this study provides valuable insights through a systematic bioinformatics analysis, identifying potential key genes associated with ASD and exploring their molecular mechanisms, making it of significant academic value. Despite some limitations in experimental validation, the innovation and proposed research directions offer valuable references for future studies. It is recommended that the authors further improve the manuscript based on the suggestions

1.The specific processing and analysis process of the datasets GSE236993 and GSE85417 mentioned in the article is somewhat brief. It is suggested to provide a detailed description of each step of data processing in the supplementary materials to increase the transparency of the methods.

2.The methods section is not detailed enough, and the statistical analysis section lacks.

3.All the abbreviations should be explained when used the first time in the manuscript. In addition, if you can avoid any of the abbreviations, it is preferred to write only full text.

4.It will be more relevant if the authors can verify their key observations in vivo and in vitro with their own samples.

5.The discussion section is relatively brief, and it is suggested to have a more detailed discussion on the functions of each key gene, especially how they play a role in the Notch signaling pathway and neural development. In addition, further exploration should be conducted to address the limitations of current research, especially in areas where data sources and validation are insufficient, and to clearly identify which aspects need to be further supplemented in future research.

6.The authors should elaborate the difference between the current manuscript and previous similar published papers, and explain the advantages of current study.

7.Multiple grammar mistakes. English needs to be improved.

**Do you want your identity to be public for this peer review?** For information about this choice, including consent withdrawal, please see our Privacy Policy

Reviewer #1: No

Reviewer #2: No

---

## [Author Response · Author response to Decision Letter 1]

13 May 2025

We sincerely thank the editor and reviewers for their insightful and constructive comments. We have carefully revised the manuscript to address all the points raised. A point-by-point response to each reviewer’s comment is provided below, with corresponding changes highlighted in the revised manuscript. All newly added or revised content is marked in red font for clarity.

We believe these revisions have significantly improved the clarity, rigor, and scientific value of our study. We appreciate your time and consideration, and we hope the revised manuscript meets the journal’s publication standards.

Thank you again for the opportunity to improve our work.

---

## [Decision Letter · Decision Letter 1]

Identification of Therapeutic Targets in Autism Spectrum Disorder through CHD8-Notch Pathway Interaction Analysis

PONE-D-25-15242R1

Dear Dr. bao,

We’re pleased to inform you that your manuscript has been judged scientifically suitable for publication and will be formally accepted for publication once it meets all outstanding technical requirements.

Kind regards,

Jinhui Liu

Academic Editor

PLOS ONE

Additional Editor Comments (optional):

I think this manuscript was well organized and it could be accepted.

Reviewers' comments:

Reviewer's Responses to Questions

**Comments to the Author**

Reviewer #1: All comments have been addressed

Reviewer #2: All comments have been addressed

2. Is the manuscript technically sound, and do the data support the conclusions?

Reviewer #1: Yes

Reviewer #2: Yes

3. Has the statistical analysis been performed appropriately and rigorously?

Reviewer #1: Yes

Reviewer #2: Yes

4. Have the authors made all data underlying the findings in their manuscript fully available?

Reviewer #1: Yes

Reviewer #2: Yes

5. Is the manuscript presented in an intelligible fashion and written in standard English?

Reviewer #1: Yes

Reviewer #2: Yes

Reviewer #1: (No Response)

Reviewer #2: The author has answered all the questions and this article is suitable for publication. I have no other questions

**Do you want your identity to be public for this peer review?** For information about this choice, including consent withdrawal, please see our Privacy Policy

Reviewer #1: No

Reviewer #2: No

---

## [Editor Report · Acceptance letter]

PONE-D-25-15242R1

PLOS ONE

Dear Dr. bao,

I'm pleased to inform you that your manuscript has been deemed suitable for publication in PLOS ONE. Congratulations! Your manuscript is now being handed over to our production team.

Kind regards,

on behalf of

Dr. Jinhui Liu

Academic Editor

PLOS ONE